# Treatment Strategy in Human Ocular Toxoplasmosis: Why Antibiotics Have Failed

**DOI:** 10.3390/jcm10051090

**Published:** 2021-03-05

**Authors:** Justus G. Garweg, Uwe Pleyer

**Affiliations:** 1Swiss Eye Institute, Rotkreuz, and Uveitis Clinic, Berner Augenklinik am Lindenhofspital, 3012 Bern, Switzerland; 2Department of Ophthalmology, Inselspital, University of Bern, 3012 Bern, Switzerland; 3Department of Ophthalmology, Campus Virchow, Charité—Universitätsmedizin Berlin, Corporate Member of Freie Universität Berlin, Humboldt-Universität zu Berlin and Berlin Institute of Health, Charité, 13353 Berlin, Germany; uwe.pleyer@chariete.de

**Keywords:** antibiotics, corticosteroids, ocular toxoplasmosis, recurrence, treatment outcomes, recurrence prophylaxis, trimethoprim-sulfamethoxazole

## Abstract

Background: There is currently no clear evidence of the effectiveness of antibiotic therapy in acute ocular toxoplasmosis (OT), but its effect as a secondary prophylaxis is undisputed. The majority of uveitis specialists advocate treatment. This meta-analytic review aims to critically analyze determinants of treatment success and to update current treatment strategies for OT in order to explain this discrepancy. Methods: A systematic literature search was performed in NCBI/PubMed, Clinical Trials, Google Scholar and ScienceDirect to retrieve pro- and retrospective studies using the key terms “ocular toxoplasmosis” or “retinochoroiditis” and “immunocompetent” and “treatment” or “therapy” and “human.” Of these, larger case series and prospective clinical studies and cross references identified from meta-analyses were selected by a manual search, and primary and secondary outcome parameters were extracted. Results: Ten case series and clinical trials reported success parameters for treatment outcomes, and four additional for recurrence prophylaxis. Five treatment studies were randomized clinical trials, three comparative and two noncomparative case series. Though several outcome parameters were reported, five of them defined time to healing, four visual gain and one lesion size as primary and secondary outcome parameters, recurrence rate as a secondary outcome parameter was reported once. No conclusive evidence was found for an antibiotic treatment effect. Four prophylaxis studies addressed the prevention of recurrences after treatment. The primary outcome in all studies was the effect of treatment and prophylaxis on recurrences, and all four found a significant effect on the risk of and time to recurrences. Conclusions: Antibiotic treatment of OT aims at controlling parasite proliferation. The absence of an effect on visual acuity and time to healing is thus not surprising. The fact that time to and number of recurrences respond to recurrence of prophylaxis proves the antibiotic effect on parasite activity.

## 1. Introduction

Toxoplasmic retinochoroiditis or ocular toxoplasmosis (OT) is the main cause of infectious posterior uveitis in several geographical areas [1,2]. The capability of the parasite to manipulate the host’s immune response influences the initial lesion presentation as well as the risk of further relapses, whereas the severity of an episode is related to the parasite’s genotype and the host immune status [3,4]. Acquired infection may be more prevalent in clinical disease than congenital, which is contrary to previous beliefs. This implies that preventing the disease would require a prevention strategy aimed not only towards pregnant women but towards the general population [3,4,5,6].

The epidemiology of OT is barely understood given the fact that 30% of the world population is chronically infected with the parasite *Toxoplasma gondii* (*T. gondii*), with a wide variability in clinical manifestations [2,7]. It is estimated that in Europe and Northern America, 2% of infected individuals will present active lesions or scars compatible with OT [2,4,8]. Thus, about 3 million individuals in Europe suffer active OT or carry corresponding scars, in many instances in the absence of a known history of OT. Clinical disease severity is usually mild to moderate and driven by a balance between parasite virulence [9,10] and host immunity [11,12].

In the last 1–2 decades, the prevalence of systemic T. gondii infection in Europe has been reported to decline by several authors [13,14,15] and to have declined and stabilized at a low level in the last decade in North America [16,17,18], whereas a still high seropositivity was reported from Germany [19]. On the other hand, the incidence of ocular disease has remained stable in recent decades in the old world [3] with 1–2% of seropositive individuals developing ocular lesions [20,21], resulting in an annual incidence of between 2200 [22] and almost 4800 symptomatic cases of OT requiring treatment in the USA [21]. Prevalence data for OT are scarce; a recent study from West Germany reported in a random sample of 12,782 individuals aged 35–74 years a prevalence of scars compatible with OT of 0.2% [23]. According to a European meta-analysis, an incidence of OT of 9.4% has to be expected based on the analysis of 24,126 patients with uveitis from 12 European countries [24], whereas the incidence of OT in seropositive patients from South America may reach 20% [20].

A high rate of *T. gondii* seropositivity in pregnant women at global, regional and country levels was reported to be associated with a high risk of maternal and congenital toxoplasmosis [25]. A much higher incidence and a more severe affection of OT has been reported from South America over the last two decades [7,8,26,27,28]. This might be at least partially related to an increasing awareness of this disease since the 1990s [29]. Based on the aforementioned decline in T. gondii seroprevalence in Europe, a significant reduction in the burden of disease has been predicted, but as yet not precipitated in the real world [17].

A lower seroprevalence in the group aged 18–49 years is associated with a higher number of individuals at risk for seroconversion in general, and particularly during pregnancy. This, however, implies an increased risk of vertical transmission to the unborn child, as seroprevalence data from Germany indicate. The seroprevalence in Germany rose from approximately 20% within the age group of 18–29 years to 77% in the 70–79 years age group. This translates into an expected annual number of almost 6400 seroconversions during pregnancy in seronegative women aged 18–49 years or 1.3% of all pregnancies in Germany, a minority of which are timely diagnosed. Even if asymptomatic, the corresponding offspring may still serve as an important source of OT during later life. Socioeconomic factors and eating behavior, male gender, keeping cats and body mass index (BMI) ≥30 may be the most prevalent independent risk factors [17,19,30]. Congenital toxoplasmosis asymptomatic at birth may thus significantly contribute to the number of ocular, and particularly macular, manifestations and T. gondii-associated vision loss [31,32,33].

Evidence-based data on the efficacy of antiparasitic drugs for the treatment of acute episodes of OT are scarce [34,35,36]. Since there are no randomized studies and the parasite cannot be effectively eliminated from the body, the need for therapy of acute OT has been questioned in principle several times. Any treatment will have to reach the acute manifestation of a chronic and persisting infection resulting from reactivation upon the weakening of the host local immune response [3,37,38,39] and needs to be based on a supportable safety profile [40]. In this context and based on the literature published in the field, this meta-analysis aimed at identifying determinants of treatment success and at updating the treatment strategy for OT and its sequalae. 

While the antibiotic treatment of acute manifestation appears questionable, level 1 evidence supports its effect on the prevention of recurrences [41,42,43,44]. Since similar antibiotics are used in both situations, fundamental questions arise regarding the absence of a therapeutic in the presence of a prophylactic effect. And the central question in the therapeutic and prophylactic situation may be whether the clinical endpoints for disease monitoring have been chosen correctly.

## 2. Materials and Methods

A systematic literature search was performed on 31 May 2020, in the NCBI/PubMed database from the National Institute of Health, USA, Clinical Trials registry, Google Scholar and Elsevier ScienceDirect to identify pro- and retrospective studies retrieved by the key terms “ocular toxoplasmosis” or «”retinochoroiditis” and “immunocompetent” and “treatment” or “therapy” and “human.” Of the reviewed articles, only those including larger case series and prospective clinical studies published within the last 20 years were selected, along with cross references identified from meta-analyses in the corresponding period using a manual search. The primary and secondary outcome parameters and the corresponding results were manually extracted.

## 3. Results

Based on the above-mentioned search terms, 35 publications were identified and provided the basis for further analysis. Treatment outcome parameters (Table 1) and outcomes were reported in 10 larger case series and clinical trials (Table 2 [45,46,47,48,49,50,51,52,53,54]). Five of the 10 treatment studies referred to European and Northern American cohorts [45,46,47,48,49], two each to Iran [50,52] and Brazil [51,54] and one to Australia [53]. Two of four recurrence prophylaxis studies in immunocompetent individuals with OT came from Brazil, where the incidence and risk of recurrences are significantly higher, thus allowing us to analyze prophylactic strategies during a reasonable study period of 24 months. Both Brazilian trials demonstrated a strong preventive effect on recurrences while under antibiotic prophylaxis [41,42,43,44]. These findings are supported by two European retrospective case series (Table 3 and Table 4 [41,42,43,44,55,56]).

Five of the treatment studies were randomized controlled trials (RCTs), three retrospective comparative case series and two noncomparative case series comparing outcomes to published evidence. The majority of the treatment studies assessed several parameters for treatment success. Five of them defined time to healing, four best-corrected visual acuity (BCVA) and one lesion size as primary outcome parameters, both of which also represented the most frequently reported secondary outcome parameters. Only one study [48] introduced recurrences as an outcome parameter (Table 2). Based on these outcome parameters, no conclusive evidence indicated an antibiotic effect.

Of the four studies addressing prophylaxis of recurrences after AB treatment, two were RCTs, and two were retrospective observational case series. The primary outcome in all studies was the effect of treatment and prophylaxis on recurrences, and all four, despite also assessing other success parameters (Table 3), reported a treatment effect on the risk of and time to recurrences (Table 4).

Strengths and weaknesses of the studies are reported in Table 1 and Table 3. Both of the first studies were RCTs, which assessed the effect of antibiotics (ABs) alone or in combination with corticosteroids on BCVA and time to healing. The first study included eyes with active and inactive anterior and posterior uveitis of possible and suspected toxoplasmic origin and reported no effect of ABs (Pyrimethamine) on BCVA, but a shorter time to healing compared to placebo [45]. In line with this, the second study found no effect of ABs (Pyrimethamina and Sulfadiazine (PY/SA)) and corticosteroids on BCVA but on time to healing and recurrences compared to steroids alone [46]. The third study refers to a non-comparative case series treated with ABs (PY/SA) and corticosteroids and reported in the absence of a comparator a positive effect on BCVA and time to healing [47], which was confirmed by a British retrospective comparative case series that reported a shorter time to healing and less recurrences after 2 different AB treatments (Pyrimethamine and Spiramycin) compared to nothing or corticosteroids alone [48], and an Australian study using different antibiotics in combination with corticosteroids [53]. A French RCT compared systemic (PY/SA) and parabulbar antibiotics (Clindamycin) and found no difference in BCVA and time to healing [49]. In two Iranian RCTs, different systemic AB regimens were compared (PY/SA vs. Trimethoprim and Sulfamethoxazole (TMP/SMZ) [50] and Azithromycin vs. TMP/SMZ [52]) which revealed no difference between the antibiotics in use and reduction in lesion size and BCVA. This was also reported from a large retrospective Brazilian case series [54]. In the absence of a comparator, a small non-comparative Brazilian case series reported a beneficial effect of ABs and steroids on time to healing and BCVA [51]. In summary, the superiority of AB treatment compared to not using any ABs was only addressed by two RCTs and one case series. One RCT included all forms of active and inactive anterior and posterior uveitis of possible toxoplasmic origin [45], the other including 20 patients with OT [46]. In the first, the number of cases with active OT was likely underrepresented compared to inactive instances; the second was with ten patients per group clearly underpowered to allow any conclusion with respect to a therapeutic effect of ABs. The positive effect of AB treatment on BCVA and recurrences compared to no treatment or corticosteroids alone of a retrospective case series [48] is the only currently available evidence in favor of AB treatment. This is surprising given the potential impact of OT on the quality of life of the affected individuals.

Compared to the lack of qualified studies for AB treatment in OT, its preventive effect on the development of recurrences is well supported by two RCTs from Brazil [41,42,43,44,45] with a follow up of 5–10 years, indicating that the prophylactic effect is limited to maximally one year after termination of the prophylaxis, which was also reported by two European case series [55,56].

In the studies reported above, limited evidence further supports a synergistic effect of the combination of systemic and intravitreal [47,48,53] ABs and corticosteroids [51] regarding time to healing, whereas one small study did not support a synergistic effect [46].

## 4. Discussion

### 4.1. General Aspects Affecting Treatment of OT

There is no doubt that AB treatment for toxoplasmosis has been shown to be effective in vitro and in animal models. Clinical evidence for this tenet was reported from a Brazilian study following patients for up to 28 years after a recently attracted systemic *Toxoplasma* infection. In total, 9.9% of the patients showed uveitis activity at diagnosis, but no retinochoroidal lesion. Antiparasitic treatment was associated with significantly less ocular involvement in this longitudinal case series. Among patients without ocular involvement at baseline, the incidence of necrotizing retinochoroiditis was 6.4/100 patient years, indicating a significant risk for the development of OT and thus likely justifying AB therapy not only for the treatment but also prevention of OT [57].

The number of uveitis specialists, who treat all patients with active OT independently of the severity and location of the disease, increased in the USA from 6 to 15% between 1991 and 2002 [58,59], whereas in Germany, 45% of uveitis specialists treat all active cases [60] compared to 62.1% in India [61] and 67.9% in Brazil [62]. Although the opinion about the need to treat varies greatly among uveitis experts, there seems to exist a generally accepted first choice treatment in the aforementioned surveys: the combination of pyrimethamine and sulfadiazine (PY/SA) with or without systemic corticosteroids seems to represent today’s gold standard for treating systemic as well as OT, whereas there exists a clear trend away from this side-effect-affected combination therapy to the fixed combination therapy with trimethoprim 160 mg and sulfamethoxazole 800 mg (TMP/SMZ) [34], which demonstrates a similar clinical effect as PY/SA, but has—in the absence of a sulfonamide allergy—an excellent safety profile [63]. Though several other regimens are in use [8,60,61,62], none seem to be superior to PY/SA [34,63]. These include, in addition to TMP/SMZ, the combination of pyrimethamine with clindamycin, atovaquone, clarithromycin or azithromycin as well as monotherapies with atovaquone or azithromycin [34,63]. It has to be kept in mind that none of the existing regimens is active against the latent stage of the infection, nor do they eradicate the infection [64]. A successful treatment will bring the infection to a dormant state without clearing the body of the parasite, which explains the inherent risk of recurrences [65,66].

Whereas most published studies have introduced change in visual acuity and time to healing as the primary treatment outcome parameter (Table 1), other parameters, such as the regression in lesion size, time to and risk of recurrences, have been considered secondary outcome parameters. The latter is surprising, since most patients become symptomatic, with vitreal floaters as the major visual complaint. Whereas visual acuity may completely recover upon the resolution of vitreal infiltration in instances not affecting the macula, the opposite holds true in any lesion affecting the fovea due to the neuroretinal tissue destruction independent of the strength of any antiparasitic treatment [67]. Whereas in three quarters of European cases, visual acuity is near-normal to normal, one quarter of patients will experience a moderately to severely reduced vision. A normal visual field in contrast is encountered in only 6% of instances, whereas 29% present a mild, 46% a moderate and 19% a severe impairment of the visual field, depending on the anatomical location of the retinal lesions and scars. The more centrally these are located, the larger the visual field defects are expected to be, resulting from a complete destruction of the nerve fiber layer in the corresponding retinal sector [68]. Taken together, visual acuity improvement and time to healing as the primary functional outcome parameters, though used in a majority of treatment studies, are not reasonably established success parameters to establish antibiotic treatment effects on parasite proliferation.

The most recent Cochrane Database Systematic Review update in 2016, for example, identified a total of four trials of sufficient quality, including 268 participants. The primary outcomes were visual acuity at least 3 months after treatment and the risk of recurrent retinochoroiditis; secondary outcomes included improvement in symptoms and signs of intraocular inflammation, changes in lesion size and adverse events [34]. In this meta-analysis, a similar change in visual acuity was found in treated and untreated eyes with a mean difference of −1.0 letters (93 observations; 95% confidence interval ranging from −7.9 to +5.9 letters) providing low-quality evidence against the effect of systemic antibiotics. Moderate-quality evidence, on the other hand, reported that treatment with antibiotics is associated with a reduced risk for recurrences (227 observations; relative risk (RR) 0.26; 95% confidence interval (CI) 0.11 to 0.63). The relevance of these results, however, was questioned, since results were similar for acute and chronic retinochoroiditis. Low-quality evidence was found for an improvement in intraocular inflammation (29 observations; RR 1.76; 95% CI 0.98 to 3.19). If participants were treated simultaneously with antibiotics and corticosteroids, almost no intraocular inflammation was reported. This indicates that the control of inflammation has a higher impact on clinical outcomes than the treatment of the underlying infectious activity. A decrease in hemoglobin, leucocyte and platelet counts; nausea; loss of appetite; rash; and arthralgia indicate an increased risk of adverse events in treated participants, which would seem not to favor treatment. [34]. The results of this meta-analysis as well as a second one including 44 treatment trials [69] demonstrate that antibiotics alone do not improve visual acuity in immunocompetent individuals, whereas a combination with corticosteroids may enhance visual recovery and inflammatory response. A small effect of antibiotic therapy on recurrence behavior was observed [34].

### 4.2. Adverse Effects of Antiparasitic Therapy

First of all, it must be emphasized that AB therapy for the treatment of OT requires a relatively long treatment time of 4–6 weeks with antibiotics that are typically given for 7–10 days in bacterial infections. This may contribute to the relatively high rate of complications in the treatment of the disease [34].

A relevant adverse event (AE) profile of pyrimethamine-based therapies in toxoplasmosis was also reported by a recent systematic review, which included 31 prospective, retrospective, observational and cohort studies and a total of 2975 patients, 13 of which (929 participants) referred to congenital toxoplasmosis (CT), 11 to ocular toxoplasmosis (n = 1284) and seven to toxoplasmic encephalitis (TE; *n* = 687). In up to 37% of patients, AE-related treatment discontinuation and/or change of therapy was reported in more than 55% of the included studies [70]. Side-effect incidences of up to 100% were reported for OT, 57.1% for TE and 61.5% for CT in the different studies. The most frequently reported AEs included bone marrow suppression with a prevalence of ≤9.0% in OT, ≤42.7% in TE and ≤50% in CT. Dermatologic side effects were reported in up to 100%, ≤10.7% and ≤10.8% as well as gastrointestinal (GI) ones in ≤11.1%, ≤17.9% and ≤2.1% in OT, TE and CT, respectively. Stevens–Johnson syndrome, as the most severe and potentially life-threatening allergic manifestation towards sulfonamides, was reported in a total of three patients (incidence 1:1000; 2 × OT; 1 × TE [70]). Hematologic AEs were reported across all manifestations, indicating the importance of blood monitoring in pyrimethamine-based regimens. The up to fivefold differences in GI and dermatological AEs between the different diagnoses are, according to the authors, explained by the absence of a systematic report and differences in the duration of treatment: getting used to treatment, particularly in CT, might possibly explain some of the differences [70]. Compared to PY/SA, treatment with TMP/SMZ seems to be a reasonable alternative treatment of OT in immunocompetent patients, particularly since this is associated with a significantly better side-effect profile, as previously mentioned [63,71].

### 4.3. Intravitreal Treatment of OT

Compared to the classical antibiotic combination treatment with PY/SA, clindamycin given intravitreally in one to three injections of 1 mg in combination with 400 µg of dexamethasone in 2-week intervals had a similar efficacy regarding the time to resolution of an active lesion, visual recovery and, interestingly, recurrence rates (12.5% compared to 14.7% within two years; *p* = 0.54), and intravitreal clindamycin therapy did not evoke any systemic toxicity compared to 6% under PY/SA combination therapy [36,72,73]. It must, however, be kept in mind that OT is deemed the local manifestation of a systemic disease, and a local treatment may not control extraocular disease activity, which accounts particularly for the risk of second eye involvement if scars are present [73]. Intravitreal steroids must, however, not be used without antibiotic coverage in order to prevent the risk of retinal necrosis in response to a profound intraocular immunosuppression [74].

### 4.4. The Role of Corticosteroids as Adjuvant Therapy for OT

A recent meta-analysis searched randomized and quasi RCTs, including immunocompetent participants of any age with active OT, comparing antiparasitic therapy plus corticosteroids versus antiparasitic therapy alone, but accepting different doses and times of initiation of corticosteroids. The authors summarized that no trial assessed the effect of adding corticosteroids to the antiparasitic therapy in active disease, despite their widespread use in clinical practice. Consequently, several questions remain to be addressed, among which is whether the use of corticosteroids as an adjunctive therapy is more effective than antiparasitic therapy alone. The other question that remains is whether corticosteroids should be initiated in parallel with the antiparasitic treatment or later in the course of treatment, and what dosage and duration of corticosteroid therapy are needed [75,76].

### 4.5. Risk of and Time to Recurrences

Recurrences of OT expose the eye to the risk of a permanent functional damage. The risk for recurrences seems related to geographic and genetic parasite and host factors. The effect of treatment on the time to recurrences has, as outlined above, not been definitively established [41,77,78,79,80]. According to a European retrospective analysis, younger patients carry a higher risk of developing a recurrence than older ones, and after each episode, 55–60% of all patients will develop a recurrence of their OT. Questions about the recurrence behavior in OT are a major concern for both patients and ophthalmologists [80]. Though not very strong, there exists some evidence for a host genetic factor determining recurrences of OT. An increased expression of interleukin 17A (IL-17A) in consequence of NOD2 gene upregulation has been reported [81]. The CD40-driven upregulation of Beclin-1 in the microglia triggers autophagy of parasites [82]. Finally, during chronic, inactive T. gondii infection within the eye, the presence of specific T-cells is required to control parasite replication. This has been correlated with a changed intraocular environment, such as cytokine (Interleukin-9) and chemokine (CXCL10) expression [39,83].

As previously mentioned, the risk of recurrences seems to be influenced by therapy [34,84]. A few years ago, a German registry reported 0.29 recurrences per year or a recurrence-free survival time of 2.5 years [84]; moreover, the risk of recurrences seems to decline with the time since last activity and patient age [80]. An effect of different antibiotic regimens on the recurrence-free survival was not found with 3.0 years without and with *T. gondii*-specific antibiotic treatment, whereas this interval dropped to 0.9 years if systemic corticosteroid monotherapy was used without antibiotics (*p* < 0.006) [55].

### 4.6. Trimethoprim-Sulfamethoxazole and the Risk of Recurrences of OT

In contrast to a lack of evidence in favor of an effect of antiparasitic treatment in active OT, its effect on recurrences seems to be supported by level 1 evidence [85]: two prospective randomized clinical trials (RCTs) found that the combination of TMP/SMZ (800/160 mg) following therapy of an active lesion is able to significantly reduce the risk of recurrences of OT. In the first study, patients were given TMP/SMZ every third day for up to 20 months [41,42]. Recurrences developed in four (6.6%) treated patients and in 15 (23.8%) controls (*p* = 0.01) [41]. After 10 years, the recurrence rate was the same in both groups (37–38%), suggesting that the prophylactic treatment effect disappears when the prophylaxis is stopped [42]. Whereas 19 out of 69 patients (27.5%) in the second RCT under placebo developed one or more recurrences during the follow-up of 6 years, only one patient (1.4%) of the prophylaxis group developed a recurrence [43,44]. In the treatment group, no case of multiple recurrences in the same individual and no treatment-limiting toxicity or other relevant side effects were recorded. Generally, recurrences were more frequent among female participants. Current evidence indicates that TMP/SMX may be used safely for prophylaxis of recurrent OT, and shows long-term benefits at least in individuals with frequent recurrences [43,44]. Obviously, prophylaxis with TMP/SMZ every third day does not completely prevent recurrences and does not have an effect beyond treatment cessation, whereas a prophylaxis given every second day almost completely suppresses recurrences over 5 years. One might speculate that a complete suppression of parasite proliferation over at least 1 year contributes to a dormant state of disease in immunocompetent individuals over 5 or more years.

Not surprisingly, 85% of Brazilian uveitis experts use prophylactic antibiotic strategies in cases with frequent recurrences [62]. In our own retrospective analysis of European patients with OT, we found a recurrence rate of 63% over 3 years, with the majority of cases (54%) presenting during the first 12 months after the last lesion activity [80], which was confirmed in another study [84]. In another retrospective study, 352 patients received a center-specific, relatively complex active-disease treatment with pyrimethamine/sulfadoxine (P/S, Fansidar^®^) 25 mg/500 mg daily for 21 days with a double-loading dose for the first 2 days, supplemented by prednisone at a starting dose of 40 mg, spiramycine 3 × 3 million international units daily for 10 days followed by azithromycin 500 mg once daily for another 6 days. Thereafter, they received a recurrence prophylaxis with a P/S 25 mg/500 mg tablet twice a week for 6 months. Under this protocol, a 3-year recurrence-free survival was observed in 90.9% of patients, the risk for recurrences peaking approximately 3.5 years after the first treatment. That the risk for recurrence was 2.82 times higher in patients with retinal scars may indicate a genetic disposition or a specific parasite–host interaction in recurrent OT [56].

Taken together, these data provide strong evidence for the preventive effect of a recurrence prophylaxis during the first 12 months after an active episode of OT, which should be considered at least in patients with an increased risk of recurrence or a high risk of severe functional impact in the case of recurrences [69,85], whereas the need of a recurrence prophylaxis in the context of intraocular surgery is still a matter of debate [86].

### 4.7. Why Antibiotics Have Failed in OT

One might expect that antibiotic treatment for OT aims at controlling parasite proliferation, as has been shown in animal models. This, however, is not the case in the publications evaluating treatment for human OT. That an effect on BCVA and time to healing was not observed is thus not surprising. That time to and number of recurrences respond to recurrence prophylaxis proves the antibiotic effect on parasite activity [41,42,43,44,55,56] while host genetic and immune factors may contribute to this risk [56,81,82,83].

In general, parameters focusing on a local infectious disease activity have to be differentiated from those affecting the immune response to this infection. In the case of OT, the first group includes the number of new lesions and the active lesion area at the time of diagnosis, as well as the number of and time to recurrences. Beyond the second group, the severity of vitreal infiltration and retinal vasculitis, time to clearance of vitreal inflammatory cells, time to scarring and regression in lesion size have to be taken into account, whereas host immune factors also may contribute to time to recurrence and number of recurrences. Consequently, host immunocompetence and the use of anti-inflammatory agents would be expected to impact second group parameters, as supported by many of the aforementioned studies.

Whereas most of the reported studies differ as much in the treatment protocol and follow-up as in their aims and primary outcomes, visual outcome and reduction in lesion size do not qualify in assessing the effect of antibiotic treatment and thus are not suitable clinical endpoints [45,46,47,48,49,50,51,52,53,54]. In consequence of this mismatch in outcomes, a report by the American Academy of Ophthalmology found that there is a lack of level I evidence to support the efficacy of routine antibiotic or corticosteroid treatment for acute OT in immunocompetent patients [36]. This must, however, not be misinterpreted as evidence for the absence of an antibiotic effect on parasite activity. As concluded by SM. Barb and colleagues [87], the literature is limited by the various outcomes and endpoints used to compare different treatment regimens for toxoplasmic retinochoroiditis. Little attention has been paid to defining treatment success other than comparisons between treatments. Five years have passed since then, while the doubts regarding the risk–benefit ratio of treatment have remained.

On the other hand, significant advances have been made in recent years through imaging methods, in particular, optical coherence tomography (OCT) and OCT angiography (OCT-A). This allows the application of morphological criteria for lesion healing, particularly to exactly measure the size of the lesions over time, which are often beyond the clinically discernible level. In addition, vitreous haze and other parameters indicating the activity of OT can be monitored. Together, they allow a better quantification of the retinal and in particular choroidal changes. Beyond their suitability for substantiate treatment effects, OCT/OCT-A findings may also contribute substantially to therapeutic decisions [88,89,90,91]. Antibiotic treatments aim at controlling parasite proliferation, which cannot readily be quantified in the clinical setting and has to be differentiated from the immune response, which is driving the severity of inflammation, time to recovery and lesion size reduction. Visual outcome, at the end, is linked to the localization of the lesion(s) but—as sufficiently demonstrated in the literature—is not controlled by any AB or anti-inflammatory treatment protocol.

The reluctance to offer AB treatment in active OT may be linked to the absence of an impact of treatment on visual outcomes [92,93] and a potentially relevant side-effect profile of the classical combination therapy with PY/SA [70]. The absence of an AB effect on visual recovery is not surprising, given the fact that ABs act on proliferating germs, but not the visual system. An effect of ABs has been reported when given in combination with corticosteroids. In this case, the time to clearance of a functionally relevant vitreal infiltration and thus of visual recovery is shorter, whereas lesion size regression is more pronounced compared to ABs given alone [47,48,51,53]. This is not surprising, since both vitreal infiltration and inflammatory tissue destruction are biomarkers of the host immune response, but only to a minor part linked to parasite–host interaction in immunocompetent individuals [3]. Whereas the effect of corticosteroids on inflammatory activity has been well established, their role in combination with antiparasitic treatment in parasite replication and reactivation is less supported by evidence; the use of corticosteroids alone, however, seems to be associated with a shorter time to recurrence [84]. To summarize the aforementioned evidence, as a result of a mis-selection of the primary outcome parameters, an effect of antiparasitic treatment has been widely neglected in the absence of a change in visual acuity. Recurrence prophylaxis studies, in contrast, have successfully confirmed the AB effect by choosing not to use BCVA, time to healing or lesion size as primary parameters for antiparasitic treatment success.

### 4.8. Therapeutic Decision Making in OT in the Absence of Evidence

Evidence is poor to moderate only, but it is generally accepted that central lesions exposing the macula to frequent recurrences deserve treatment in favor of safety first, if supported by the patient. A strong inflammatory response is associated with vision loss, the restoration of which requires the addition of a corticosteroid treatment. As the clinical course of disease, i.e., the severity of inflammatory infiltration and the frequency of recurrences, is more aggressive in South American individuals, a systematic antiparasitic treatment should be considered in individuals from this region of the world. Systemic immune defects and iatrogenic immunosuppression, caused by either the use of intraocular corticosteroids or profound and sustained systemic immunosuppression, e.g., after organ transplantation or in the context of inflammatory bowel disease, should trigger a recurrence prophylaxis in individuals after treatment of an active retinochoroidal lesion suggestive of OT. There exists no clear guidance with respect to OT in patients with one seeing eye only. In these cases, if the central vision is not affected, a treatment may be discussed, whereas an active lesion within the central 30–40° should be treated not only to preserve visual acuity but also to prevent clinically relevant visual field defects. A negative effect of any recurrence on the visual field is more likely than that on BCVA. Based on published evidence, therapy is not generally advised for small lesions in the periphery, whereas any high-dose systemic and any intravitreal steroid therapy in an eye with ocular lesions resembling OT should trigger AB prophylaxis [74,87,93].

## 5. Conclusions

Treatment decisions are currently left to the conviction of the treating physician and her or his informed patients [34,69,93]. Generally accepted clinical endpoints for the treatment of active and recurrent OT have yet to be established, whereas visual acuity and time to healing or resolution of inflammatory vitreal changes, the most frequently applied primary outcome parameters, have proven useless for testing antibiotic efficacy in ocular toxoplasmosis. Based on this, the absence of evidence for an effect of the treatment of OT is not evidence of a lack of effect, as is frequently misunderstood. Antiparasitic treatment should, when in doubt, be given if supported by the patients with the aim of reducing the risk of recurrences after treatment (level 3 evidence). Intravitreal clindamycin in combination with dexamethasone is as effective as systemic AB treatment (level 2 evidence). Trimethoprim-sulfamethoxazole seems to be as effective as pyrimethamine and sulfadiazine, but with a significantly improved safety profile (level 2 evidence), whereas intermittent trimethoprim-sulfamethoxazole prophylaxis over 12 months is capable of preventing recurrences of the disease (level 1 evidence). Whereas according to few case reports, the only use of corticosteroids, namely, given intravitreally, may have disastrous consequences, the addition of corticosteroids seems to enhance the time to lesion healing (level 3 evidence).

## Figures and Tables

**Table 1 jcm-10-01090-t001:** Treatment of active ocular toxoplasmosis (OT). Overview over registered primary and secondary outcome parameters.

Reference Number	First Author	Year of Publication	*n* (Cases)	Country of Origin	Treatment	BCVA	Recurrence Events	Time to Recurrence	Lesion Size	Change in Inflammation	Time to Healing	Safety and Side Effects	Remarks
[45]	Perkins ES	1956	164	Great Britain	PY vs. placebo	+	−	−	−	+	−	+	comparison of dye-test positive and negative uveitis, toxoplasmosis in ~25%
[46]	Acers TE	1964	20	USA	PY/SA/steroids vs. steroids alone	+	+	−	−	+	+	+	active and inactive lesions included
[47]	Ghosh M	1965	114	USA	PY/SA/steroids	+	+	−	−	−	+	+	lesions in 76% within 11 weeks inactive
[48]	Nolan J	1968	69	Great Britain	PY vs. Spiramycin vs. nothing + steroids	−	+	−	−	−	+	−	results indicate AB effect
[49]	Colin J	1989	29	France	PY/SA vs. Clindamycin subconjunctivally	+	+	−	−	−	+	+	subconjunctival clindamycin as effective as systemic PY/SA, but no untreated control group
[50]	Soheilian M	2005	59	Iran	PY/SA vs. TMP/SMZ	+	+	−	+	−	−	+	reduction in lesion size and improvement in VA comparable between TMP/SMZ and PY/SA
[51]	Zamora YF	2015	16	Brazil	intravitreal clindamycin + dexamethasone	+	−	−	−	−	+	−	five eyes were systemically pre-treated, no control
[52]	Lashay A	2016	27	Iran	Azithromycin vs. TMP/SMZ for 6–12 weeks	+	−	−	+	+	+	−	reduction in lesion size and improvement in VA comparable between TMP/SMZ and azithromycin
[53]	Yates WB	2019	48	Australia	different treatments; Clindamycin in 71%, steroids after 1 week, PY/SA, if macula at risk	+	+	−	+	−	−	−	fewer recurrences and better BCVA compared to published series
[54]	Casoy J	2020	451	Brazil	Six different AB regimens, no steroids, TMP/SMZ and PY/SA most frequent	+	−	−	−	−	+	+	all AB treatment combinations similarly effective and supportable regarding their side effects

ABs, antibiotics; BCVA, best-corrected visual acuity; PY, pyrimethamine; SA, sulfadiazine; TMP/SMZ, trimethoprim 800 mg and sulfamethoxazole 160 mg; RCT, randomized clinical trial; OT, ocular toxoplasmosis.

**Table 2 jcm-10-01090-t002:** Treatment of active OT. Treatment and primary study outcomes.

Reference Number	First Author	Year of Publication	Design	Country of Origin	*n* (Cases)	Follow Up (Months)	Treatment	Primary Outcome	AB Effect on Primary Outcome
[45]	Perkins ES	1956	RCT	Great Britain	164	1	PY vs. placebo	Effect of ABs on BCVA and uveitis	no effect of ABs on BCVA, but shorter time to healing compared to placebo. Active AND inactive cases were included
[46]	Acers TE	1964	RCT	USA	20	24	PY/SA/steroids vs. steroids alone	Effect of ABs and steroids on BCVA and time to healing	no effect of AB + corticosteroids on BCVA and time to healing compared to corticosteroids alone
[47]	Ghosh M	1965	non-comparative case series	USA	114	24	PY/SA/steroids	Time to healing under ABs and steroids	effect of ABs and steroids on time to healing
[48]	Nolan J	1968	retrospective case series	Great Britain	69	108	PY vs. Spiramycin vs. nothing + steroids	Effect of ABs and steroids on time to healing and recurrences	shorter time to healing, less recurrences
[49]	Colin J	1989	RCT	France	29	14	PY/SA vs. Clindamycin subconjunctivally	Effect of ABs on BCVA	no difference between ABs on BCVA and time to healing
[50]	Soheilian M	2005	RCT	Iran	59	24	PY/SA vs. TMP/SMZ	Difference of 2 different ABs on lesion size reduction	no difference between ABs on reduction in lesion size and BCVA
[51]	Zamora YF	2015	non-comparative case series	Brazil	16	12	intravitreal clindamycin + dexamethasone	Effect of ABs on time to healing and changes in BCVA	ABs and steroids improve time to healing and BCVA
[52]	Lashay A	2016	RCT	Iran	27	3	Azithromycin vs. TMP/SMZ	Effect of ABs on time to healing	no difference of ABs on time to healing, lesion size and BCVA
[53]	Yates WB	2019	retrospectivecase series	Australia	48	26	different treatments; Clindamycin in 71%, steroids after 1 week, PY/SA, if macula at risk	Effect of ABs and steroids on BCVA	effect of ABs and steroids on BCVA and time to healing
[54]	Casoy J	2020	retrospective case series	Brazil	451	nr	6 regimens, no steroids, TMP/SMZ and PY/SA most frequent	Comparative effect of different ABs	no difference between ABs on time to healing and BCVA

ABs, antibiotics; PY, pyrimethamine; SA, sulfadiazine; TMP/SMZ, trimethoprim 800 mg and sulfamethoxazole 160 mg; RCT, randomized clinical trial; OT, ocular toxoplasmosis.

**Table 3 jcm-10-01090-t003:** Prophylaxis of OT recurrences. Overview of registered primary and secondary outcome parameters.

Reference Number	First Author	Year of Publication	*n* (Cases)	Country of Origin	Treatment	BCVA	Recurrence Events	Time to Recurrence	Lesion Size	Change in Inflammation	Time to Healing	Safety and Side Effects	Remarks
[41,42]	Silveira C	2002/15	124	Brazil	treatment and recurrence prophylaxis with TMP/SMZ 2×/week over 20 months	−	+	+	−	+	−	+	After 2 years, fewer recurrences in the prophylaxis group. After 10 years, recurrence rate identical in both groups.
[55]	Reich M	2016	84	Germany	20 different AB regimen +/− steroids	−	+	+	−	−	−	−	time to recurrence not longer after ABs than without therapy, faster recurrence after steroids without AB
[56]	Borkowski PK	2016	352	Poland	treatment and recurrence prophylaxis for 6 months with Pyrimethamine and Sulfadoxine (Fansidar^®^)	−	+	+	−	−	−	−	Pyrimethamin/Sulfadoxine treatment and prophylaxis for 6 months prevented recurrences over 3.5 years
[43,44]	Fernandes-Felix JP	2016/20	141	Brazil	treatment and recurrence prophylaxis with TMP/SMZ 3×/week for 12 months	+	+	+	−	−	−	−	Effect of TMP/SMZ treatment and prophylaxis over 12 months on recurrences over 5 years

AB, antibiotics; TMP/SMZ, trimethoprim 800 mg and sulfamethoxazole 160 mg; RCT, randomized clinical trial; OT, ocular toxoplasmosis.

**Table 4 jcm-10-01090-t004:** Prophylaxis of OT recurrences. Prophylactic protocol and primary study outcomes.

Reference Number	First Author	Year of Publication	Design	Origin	*n* (Cases)	Follow Up (Months)	Treatment	Primary Outcome	AB Effect on Primary Outcome
[41,42]	Silveira C	2002/15	RCT	Brazil	124	120	treatment and recurrence prophylaxis with TMP/SMZ 2×/week over 20 months	Effect of AB treatment and prophylaxis on recurrences	recurrences under therapy and prophylaxis for 12 months less frequent
[55]	Reich M	2016	observational case series	Germany	84	36	20 different AB regimen +/− steroids	recurrence-free survival after ABs and steroids	no difference in recurrence risk for treated and untreated OT. Recurrences more frequent under steroids without AB coverage
[56]	Borkowski PK	2016	non-comparative case series	Poland	352	42	treatment and recurrence prophylaxis for 6 months with Pyrimethamine and Sulfadoxine (Fansidar^®^)	recurrence behaviour	recurrences under therapy and prophylaxis for 6 months less frequent
[43,44]	Fernandes-Felix JP	2016/20	RCT	Brazil	141	72	treatment and recurrence prophylaxis with TMP/SMZ 3x/week for 12 months	no recurrences over 3, single over 5 years after prophylaxis	recurrences under therapy and prophylaxis for 12 months less frequent

ABs, antibiotics; TMP/SMZ, trimethoprim 800 mg and sulfamethoxazole 160 mg; RCT, randomized clinical trial; OT, ocular toxoplasmosis.

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
