# Peer review of "Treatment Strategy in Human Ocular Toxoplasmosis: Why Antibiotics Have Failed"

_jcm, 2021, doi:10.3390/jcm10051090_

Round 1
Reviewer 1 Report
The authors analyzed determinants of treatment success and updated treatment strategies for ocular toxoplasmosis.
Overall, through a systemic literature search and meta-analyses, the treatment and outcome parameters of the OT patients in clinical study and case series were well organized. Additionally, the authors appropriately presented the clinical questions of what the OT patients' treatment outcome parameters would be. This study is worthy of notice because it delivers a practical and clinical message.
Several minor comments to the authors
1. The "OT" abbreviation in the abstract should be written in its expanded form the first time.
2. The readability of Tables is poor. Increasing the font size or changing the font will help readability.
3. Some typo errors.
Author Response
Reviewer 1:
The authors analyzed determinants of treatment success and updated treatment strategies for ocular toxoplasmosis. Overall, through a systemic literature search and meta-analyses, the treatment and outcome parameters of the OT patients in clinical study and case series were well organized. Additionally, the authors appropriately presented the clinical questions of what the OT patients' treatment outcome parameters would be. This study is worthy of notice because it delivers a practical and clinical message.
Response: The authors wish to thank the reviewer for her/his generally very positive vote.
Several minor comments to the authors:
- The "OT" abbreviation in the abstract should be written in its expanded form the first time.
Response: thanks for reminding, done
- The readability of Tables is poor. Increasing the font size or changing the font will help readability.
Response: This is obviously correct. We have now added as tables in an xls file to allow the typesetting to optimally handle this issue.
- Some typo errors.
Response: we re-read the manuscript in order to remove residual spelling and linguistic errors.
Thanks for your constructive review,
JGGarweg
Reviewer 2 Report
This review describes a meta-analysis of studies on ocular toxoplasmosis treatment. The purpose is to identify the investigated parameters and the effect of different treatment regimens on them. The most important parameters were time to healing, gain of vision and, for some studies, recurrence rate. The authors then discuss different aspects of treatment regarding efficiency, side effects and modes of administration.
Despite the clinical importance, there is still no consensus of treatment, worse still, not even if to treat or not. Therefore, studies on treatments and their outcomes are extremely valuable. Very few articles so far present a synthesis of our knowledge. Therefore, this review is more than welcome to reach a broad audience. Moreover, it is outstanding regarding the number of original studies evaluated, and the detailed analysis of these studies. The discussion points out the essential problems and will give a useful guideline for ophthalmologists on the risk-benefit balance of treatment in a particular setting. The manuscript is very well written and understandable, the language excellent. I will just add a few suggestions or questions:
- Probably due to the great quantity of information in the papers cited in the tables, it is somewhat difficult to get details of what they are really saying, without consulting them. I am a bit surprised that the results section does hardly present any results, in terms of outcomes. These are presented in the discussion, but mostly without direct reference to the studies listed in the tables. So, it is often difficult to get the link between the results and the study. There is short information on the outcome within the tables, but given the complexity of some studies (20 different treatments, +/- corticosteroids etc.) sometimes difficult to interpret just with some ‘+’ or ‘-‘. I think that the results text should summarize the most salient findings, with references. Alternatively, or additionally, the findings escribed in the discussion section should be attributed to a study in the tables.
- As the situation in South America is rather different, studies conducted in such regions should be identified (I think there are some in Table 2). In general, I would have liked some more information on the treatment effects in this region. The authors state that Brazilian ophthalmologists treat more often, or that the course of disease is more aggressive, but not if treatment is more or less efficient than in Europe. When I did not miss any information, none of the studies in Table 1 treats South America, but specialists there are in particular need of data.
- I am not sure if the title reflects the results. These results show that antibiotics do not affect healing, but the possible reasons are just explained in the discussion, and are not backed by the results. Moreover, according to the results, antibiotics do seem to be useful by lowering recurrence rates.
- The beneficial effect of corticosteroids on healing is indicated on several occasions. In contrast, ‘no steroid therapy without antibiotic cover’ is often heard. To guide ophthalmologists in this decision, the separate and synergistic effects in the studies of the two could be briefly pointed out.
- When reading ‘to prevent clinicalally relevant visual field defects’ (l. 403), I have the feeling that the usefulness of treatment to prevent future lesions (recurrences) should be pointed out more prominently. I am not sur if all ophthalmologists think of future lesions when treating actual ones.
Author Response
Reviewer 2:
This review describes a meta-analysis of studies on ocular toxoplasmosis treatment. The purpose is to identify the investigated parameters and the effect of different treatment regimens on them. The most important parameters were time to healing, gain of vision and, for some studies, recurrence rate. The authors then discuss different aspects of treatment regarding efficiency, side effects and modes of administration.
Despite the clinical importance, there is still no consensus of treatment, worse still, not even if to treat or not. Therefore, studies on treatments and their outcomes are extremely valuable. Very few articles so far present a synthesis of our knowledge. Therefore, this review is more than welcome to reach a broad audience. Moreover, it is outstanding regarding the number of original studies evaluated, and the detailed analysis of these studies. The discussion points out the essential problems and will give a useful guideline for ophthalmologists on the risk-benefit balance of treatment in a particular setting. The manuscript is very well written and understandable, the language excellent. I will just add a few suggestions or questions:
Response: Thanks to the reviewer for the highly appreciated and positive review.
- Probably due to the great quantity of information in the papers cited in the tables, it is somewhat difficult to get details of what they are really saying, without consulting them. I am a bit surprised that the results section does hardly present any results, in terms of outcomes. These are presented in the discussion, but mostly without direct reference to the studies listed in the tables. So, it is often difficult to get the link between the results and the study. There is short information on the outcome within the tables, but given the complexity of some studies (20 different treatments, +/- corticosteroids etc.) sometimes difficult to interpret just with some ‘+’ or ‘-‘. I think that the results text should summarize the most salient findings, with references. Alternatively, or additionally, the findings escribed in the discussion section should be attributed to a study in the tables.
Response: Thank you for pointing on this. We adopted the text accordingly. The added text now reads: “Strengths and weakness of the studies is reported in Tables 1A and 2A. Both first studies were RCTs which assessed the effect of antibiotics (AB) alone or in combination with corticosteroids on BCVA and time to healing. The first study included eyes with active and inactive anterior and posterior uveitis of possible and suspected toxoplasmic origin and reported no effect of AB (Pyrimethamine) on BCVA, but a shorter time to healing compared to placebo [45]. In line, the second study found no effect of AB (Pyrimethamina and Sulfadiazine (PY/SA)) and corticosteroids on BCVA but on time to healing and recurrences compared to steroids alone [46]. The third study refers to a non-comparative case series treated with AB (PY/SA) and corticosteroids and reported in the absence of a comparator a positive effect on BCVA and time to healing [47] which was confirmed by a British retrospective comparative case series which reported a shorter time to healing and less recurrences after 2 different AB treatments (Pyrimethamine and Spiramycin) compared to nothing or corticosteroids alone [48], and an Australian study using different antibiotics in combination with corticosteroids [53]. A French RCT compared systemic (PY/SA) and parabulbar antibiotics (Clindamycin) and found no difference on BCVA and time to healing [49]. In two Iranian RCTs, different systemic AB regimens were compared (PY/SA vs Trimethoprim and Sulfamethoxazole (TMP/SMZ) [50] and Azithromycin vs TMP/SMZ [52]) which revealed no difference between the antibiotics in use and reduction of lesion size and BCVA. This was also reported from a large retrospective Brazilian case series [54]. In absence of a comparator, a small non-comparative Brazilian case series reported a beneficial effect of AB and steroids on time to healing and BCVA [51]. In summary, superiority of AB treatment compared to not using any AB was only addressed by two RCTs and one case series. One RCT included all forms of active and inactive anterior and posterior uveitis of possible toxoplasmic origin [45], the other including 20 patients with OT [46]. In the first, the number of cases with active OT was likely underrepresented compared to inactive instances, the second was with ten patients per group clearly underpowered to allow any conclusion with respect to a therapeutic effect of AB. The positive effect of AB treatment on BCVA and recurrences compared to no treatment or corticosteroids alone of a retrospective case series [48] is the only currently available evidence in favour of AB treatment. This is surprising given the potential impact of OT on quality of life of the affected individuals”
- As the situation in South America is rather different, studies conducted in such regions should be identified (I think there are some in Table 2). In general, I would have liked some more information on the treatment effects in this region. The authors state that Brazilian ophthalmologists treat more often, or that the course of disease is more aggressive, but not if treatment is more or less efficient than in Europe. When I did not miss any information, none of the studies in Table 1 treats South America, but specialists there are in particular need of data.
Response: We fully agree with the reviewer in this point. Here, we aimed, however, not to analyse treatment effects, but reasons for treatment failures in the mentioned studies, which are closely linked to treatment outcomes. Consequently, different treatment attitudes in Europe and Northern America compared to Southern America was not in the focus of this work. Nevertheless, we added information pertaining to the country of the study in tables 1A and 2A and elaborated this point in the Results chapter.
It is also our strong believe, the different severity and burden of disease, requiring a more aggressive treatment and prophylaxis in Southern American patients and parasite strains. Though this is beyond the focus of our study, the corresponding text was revised and now reads: “Five of the 10 treatment studies referred to European and Northern American cohorts [45-49], two each to Iran [50, 52] and Brazil [51, 54], and one to Australia [53]. Two of four recurrence prophylaxis studies in immunocompetent individuals with OT came from Brazil, where the incidence and risk of recurrences are significantly higher, thus allowing to analyze prophylactic strategies during a reasonable study period of 24 months. Both Brazilian trials demonstrated a strong preventive effect on recurrences while under antibiotic prophylaxis [41-44]. These findings are supported by two retrospective European retrospective case series (Tables 2A and 2B [41-44, 55, 56])”.
- I am not sure if the title reflects the results. These results show that antibiotics do not affect healing, but the possible reasons are just explained in the discussion, and are not backed by the results. Moreover, according to the results, antibiotics do seem to be useful by lowering recurrence rates.
Response: We did not aim at comparing treatment and prophylaxis effects, but rather wished to point out, why the discussion if to treat or not, is based on the wrong assumptions. In response to the reviewer, the title was adapted and now reads: “Treatment strategy in human ocular toxoplasmosis: Why antibiotics have failed” Moreover, the discussion was modified and pertaining information added: “No doubt, that AB treatment for toxoplasmosis has been shown effective in vitro and in animal models. Clinical evidence for this tenet was reported from a Brazilian study following patients for up to 28 years after a recently attracted systemic Toxoplasma infection. 9.9% of the patients showed uveitis activity at diagnosis, but no retinochoroidal lesion. Antiparasitic treatment was associated with significantly less ocular involvement in this longitudinal case series. Among patients without ocular involvement at baseline, the incidence of necrotizing retinochoroiditis was 6.4/100 patient years, indicating a significant risk for the development of OT and thus likely justifying AB therapy not only for the treatment, but also prevention of OT [57]”.
- The beneficial effect of corticosteroids on healing is indicated on several occasions. In contrast, ‘no steroid therapy without antibiotic cover’ is often heard. To guide ophthalmologists in this decision, the separate and synergistic effects in the studies of the two could be briefly pointed out.
Response: We agree and elaborated the synergistic effect of combining corticosteroids with antibiotics for the time to healing in Results: “In the studies reported above, limited evidence further supports a synergistic effect of the combination of systemic [47, 48, 53] and intravitreal AB and corticosteroids [51] regarding time to healing whereas one small study did not support a synergistic effect [46]”. In the conclusion, we added “Whereas according to few case reports, the only use of corticosteroids, namely given intravitreally, may have disastrous consequences, the addition of corticosteroids seems to enhance the time to lesion healing (level 3 evidence).”
- When reading ‘to prevent clinically relevant visual field defects’ (l. 403), I have the feeling that the usefulness of treatment to prevent future lesions (recurrences) should be pointed out more prominently. I am not sur if all ophthalmologists think of future lesions when treating actual ones. Response: We fully agree with the reviewer and have adopted the text in the discussion accordingly to: “Antiparasitic treatment was associated with significantly less ocular involvement in this longitudinal case series. Among patients without ocular involvement at baseline, the incidence of necrotizing retinochoroiditis was 6.4/100 patient years, indicating a significant risk for the development of OT and thus likely justifying AB therapy not only for the treatment, but also prevention of OT [57].”. In the mentioned paragraph, we added: “A negative effect of any recurrence on the visual field is more likely than that on BCVA.”
Thanks for the constructive and helpful review,
JGGarweg